# HPV Vaccination and HPV Outcomes After LEEP: A Retrospective Population-Based Cohort Study from Northern Norway, 2022–2024

**DOI:** 10.3390/vaccines14010044

**Published:** 2025-12-30

**Authors:** Sveinung Wergeland Sørbye, Mona Antonsen, Elin Richardsen

**Affiliations:** 1Department of Clinical Pathology, University Hospital of North Norway, 9006 Tromsø, Norway; mona.antonsen@unn.no (M.A.); elin.richardsen@unn.no (E.R.); 2Department of Medical Biology, Faculty of Health Sciences, UiT The Arctic University of Norway, 9019 Tromsø, Norway

**Keywords:** HPV vaccination, loop electrosurgical excision procedure (LEEP), cervical intraepithelial neoplasia, post-treatment follow-up, HPV test positivity, HPV16, HPV18, registry-based cohort, population-based study, Norway

## Abstract

**Background/Objectives**: Women treated with loop electrosurgical excision procedure (LEEP) for high-grade cervical intraepithelial neoplasia (CIN2+) remain at risk of HPV detection during follow-up. We assessed whether HPV vaccination was associated with HPV positivity at the first post-treatment follow-up after LEEP. **Methods**: This retrospective population-based cohort included women aged 20–79 years treated by LEEP in Troms and Finnmark, Norway, during 2022–2024 (*n* = 1052). Vaccination status, timing, and vaccine product were obtained from the national immunization register (SYSVAK). Follow-up HPV results (overall HPV, HPV16, HPV18, and pooled other HPV types; Roche cobas 4800 channels) were retrieved from SymPathy. **Results**: Overall, 329/1052 women (31.3%) were HPV-positive at first follow-up. HPV positivity was 37.7% (200/530) among unvaccinated women and 24.7% (129/522) among vaccinated women (ARR 13.0 percentage points; 95% CI 7.5–18.6; RR 0.655; 95% CI 0.544–0.788; *p* = 5.2 × 10^−6^). HPV16 was detected in 5.9% vs. 9.4% (*p* = 0.0335), and pooled other HPV types in 18.0% vs. 28.7% (*p* = 4.3 × 10^−5^); HPV18 did not differ (2.9% vs. 2.5%; *p* = 0.671). In adjusted analyses, vaccination in the year of LEEP was associated with lower risk of follow-up HPV positivity (aRR 0.592; 95% CI 0.444–0.789; *p* = 0.000348). **Conclusions**: HPV vaccination before the first post-treatment follow-up was associated with lower HPV positivity after LEEP. As this outcome is a surrogate endpoint and residual confounding is possible, studies with standardized follow-up and long-term clinical endpoints are needed.

## 1. Introduction

Persistent infection with oncogenic human papillomavirus (HPV) is the necessary cause of virtually all cervical cancers and is the principal driver of high-grade cervical intraepithelial neoplasia (CIN2+) [1]. Prophylactic HPV vaccination has therefore become a cornerstone of cervical cancer prevention, complementing organized screening and treatment strategies [2]. However, despite effective excisional treatment, women treated for CIN2+—most commonly by loop electrosurgical excision procedure (LEEP)—remain at increased risk of persistent HPV infection and subsequent disease during follow-up, and they require post-treatment surveillance [3].

The rationale for considering HPV vaccination around the time of excisional treatment is grounded in tertiary prevention. Although prophylactic vaccines are not therapeutic and are not expected to eradicate established HPV infection on their own, vaccination may reduce the likelihood of new infections, re-infection with the same genotype, or reactivation, and may limit the spread of virus to adjacent susceptible epithelium after removal of the primary lesion. The biological plausibility of these mechanisms and the clinical relevance of preventing recurrent disease after treatment have stimulated a rapidly expanding volume of studies on “adjuvant” HPV vaccination in women undergoing excisional therapy [4].

Several systematic reviews and meta-analyses suggest that HPV vaccination administered before or after excisional treatment is associated with lower risk of recurrent CIN2+ compared with treatment alone, but they also highlight substantial heterogeneity across studies, varying outcome definitions, differences in timing of vaccination and follow-up, and the inherent limitations of non-randomized designs [5,6,7]. In parallel, observational cohort studies, including those embedded in routine clinical pathways, have reported reductions in persistent or recurrent CIN2+ among vaccinated women, reinforcing the potential clinical benefit. Nevertheless, uncertainty persists, in part because statistically consistent findings are not always matched by a fully resolved biological explanation, and because confounding by indication and incomplete clinical covariate capture can influence estimates in real-world datasets [8].

Clinical guidance has begun to reflect this evolving evidence base. The American Society for Colposcopy and Cervical Pathology (ASCCP) recommends adherence to routine and catch-up vaccination guidelines and supports shared clinical decision-making regarding HPV vaccination in previously unvaccinated individuals aged 27–45 years undergoing treatment for CIN2+, acknowledging a possible adjuvant benefit while recognizing limitations in the certainty of evidence [9].

More recent studies have also focused on outcomes beyond histologic recurrence, including virological endpoints and the role of the 9-valent HPV vaccine. Data on 9-valent vaccination in the post-excision setting have suggested reduced risk of recurrent CIN2+ in vaccinated women, and other studies have examined HPV test outcomes after treatment as a marker of viral persistence or clearance [10,11]. In addition, analyses of clinical trial populations indicate that vaccination occurring before excisional surgery can reduce subsequent lower genital tract disease related to vaccine HPV types, supporting the concept that vaccine-induced immunity can have meaningful post-treatment implications. Together, these findings support ongoing investigation of post-treatment vaccination strategies, particularly in settings where vaccination histories can be reliably ascertained and follow-up HPV testing is routinely performed [12,13].

Population-based studies with linkage between pathology databases, follow-up HPV testing, and national immunization registries are well positioned to address key evidence gaps. In particular, real-world cohorts can evaluate vaccine uptake across age groups, compare different vaccine products used in routine practice, and assess genotype-specific HPV outcomes at post-treatment follow-up. Such evidence is especially relevant in regions with comprehensive registries, where routinely collected data enable large-scale evaluation with minimal loss to follow-up for key endpoints. In a previous cohort from Troms and Finnmark (2022), we observed that HPV vaccination after conization was associated with a higher probability of a negative follow-up HPV test [14]. Building on this regional study, the present study expands the cohort to three treatment years (2022–2024), includes women aged 20–79 years, and evaluates HPV outcomes by vaccine type.

In the present retrospective population-based cohort study from Troms and Finnmark counties in Northern Norway, we examined women treated by LEEP during 2022–2024 and linked HPV vaccination status from the national immunization register (SYSVAK) with follow-up HPV results recorded in routine clinical practice. We aimed to (i) describe HPV positivity and the distribution of HPV16, HPV18, and other HPV types at post-treatment follow-up across age groups; (ii) assess the association between HPV vaccination and HPV positivity after LEEP; and (iii) explore whether post-treatment HPV outcomes differed by vaccine product (bivalent HPV vaccine, quadrivalent HPV vaccine, or 9-valent HPV vaccine) compared with no vaccination.

## 2. Materials and Methods

### 2.1. Study Design and Setting

We conducted a retrospective, population-based cohort study of women treated by loop electrosurgical excision procedure (LEEP) in Troms and Finnmark counties, Northern Norway, during 2022–2024. The study was performed at the Department of Clinical Pathology, University Hospital of North Norway (UNN), which receives and archives cervical cytology, HPV tests, cervical biopsies, LEEP specimens, and hysterectomy specimens from the region in the laboratory information system SymPathy (Tieto, Helsinki, Finland; version 5.14.4.2).

### 2.2. Data Sources and Record Linkage

Clinical and laboratory information was retrieved from SymPathy, including procedure year, follow-up HPV results, and HPV genotype channels (HPV16, HPV18, and pooled ‘other’ HPV types as reported by the assay). The pooled ‘other’ channel does not provide individual genotype information and therefore does not allow differentiation between vaccine-covered and non-vaccine HPV types within this category. HPV vaccination data (dates and vaccine product) were obtained from the Norwegian Immunization Registry (SYSVAK, https://www.fhi.no/en/va/norwegian-immunisation-registry-sysvak/, accessed on 26 December 2025) [15]. Individuals with a valid national personal identification number have a SYSVAK record; absence of any registered HPV vaccine doses was classified as unvaccinated. Individual-level linkage between SymPathy and SYSVAK was performed using the unique national personal identification number.

### 2.3. Study Population

The source population comprised all women aged 20–79 years residing in Troms and Finnmark with a LEEP specimen processed at the Department of Clinical Pathology, University Hospital of North Norway (UNN), during 1 January 2022 to 31 December 2024 (*n* = 1110), identified in SymPathy. According to Norwegian guidelines, the first post-treatment follow-up after LEEP is recommended at approximately 6 months. At UNN, a local reminder system is used; if the recommended follow-up test is not registered within 3 months after the scheduled follow-up time, reminders are sent to the responsible gynecologist and/or general practitioner. For the present study, women were eligible for inclusion if they had (i) a registered first post-treatment follow-up HPV test result in SymPathy (coded as negative or positive) and (ii) a recorded HPV vaccination status in the national immunization register (SYSVAK). Women without a valid follow-up HPV test result were excluded (*n* = 58), including women with cervical cancer in the LEEP specimen and/or hysterectomy or radiotherapy before follow-up HPV testing (*n* = 16), women who had moved abroad (*n* = 5), and women without a registered follow-up HPV test as of November 2025 (*n* = 37; reminders were sent to the responsible gynecologist and/or general practitioner). The final analytical cohort comprised 1052 women with a valid follow-up HPV test (Figure 1). Pre-treatment HPV status was available for 1046/1052 women (99.4%).

### 2.4. HPV Testing and Outcome Definitions

HPV testing at UNN was performed using the Roche Cobas 4800 system (Roche Molecular Systems, Pleasanton, CA, USA), which reports HPV16 and HPV18 individually and provides a pooled result for twelve additional HPV types (HPV31, 33, 35, 39, 45, 51, 52, 56, 58, 59, 66, and 68).

The primary outcome was HPV positivity at the first recorded post-treatment follow-up HPV test in SymPathy (routine follow-up typically scheduled around six months after LEEP). Secondary outcomes were positivity for HPV16, HPV18, and pooled “other” HPV types, based on the assay channels.

### 2.5. Exposure Definitions: HPV Vaccination and Vaccine Type

Vaccination status was determined from SYSVAK. Women were classified as vaccinated if they had received ≥1 dose of any HPV vaccine registered in SYSVAK (regardless of number of doses), and unvaccinated if no HPV vaccine was recorded. Vaccine product was categorized as bivalent (Cervarix; GlaxoSmithKline Biologicals s.a., Rixensart, Belgium), quadrivalent (Gardasil 4; Merck Sharp & Dohme LLC, West Point, PA, USA), nonavalent (Gardasil 9; Merck Sharp & Dohme), or none, based on the registered vaccine brand in SYSVAK.

For contextual interpretation of vaccine uptake by birth cohort, we considered the Norwegian school-based HPV vaccination program introduced in 2009 for girls in 7th grade (initially using the quadrivalent vaccine; later replaced by Cervarix; from 2018 offered to both girls and boys) and the national catch-up program offered in 2016–2019 to women aged 20–25 years (primarily birth cohorts 1991–1996). Gardasil 9 is not included in the Norwegian national vaccination programs and is typically administered outside the program on prescription, paid by the individual (out of pocket). Vaccination timing in relation to treatment and follow-up was handled by defining vaccination status relative to the first post-treatment follow-up HPV test: vaccine doses registered after the follow-up HPV test were not considered to have influenced the follow-up HPV result and were therefore classified as unvaccinated for the main analyses. In this cohort, most Gardasil 9 vaccinations were registered in the same calendar year as LEEP (typically a few months before or after treatment).

### 2.6. Variables

Age at treatment was recorded as a continuous variable and categorized as 20–29, 30–39, 40–49, 50–59, 60–69, and 70–79 years. Calendar year of LEEP (2022, 2023, 2024) was used for stratified analyses. Pre-treatment HPV status (fHPV) was defined as the most recent recorded HPV test prior to LEEP (0 = negative, 1 = positive); missing values were coded as 9. HPV status after treatment (eHPV) was defined as HPV positivity at the first registered post-treatment follow-up HPV test (0 = negative, 1 = positive). Type-specific outcomes were recorded using separate indicators for HPV16, HPV18, and pooled other HPV types at follow-up.

HPV vaccination variables were derived from SYSVAK and included overall vaccination status defined relative to outcome assessment (vacc_01: ≥1 HPV vaccine dose recorded before the first follow-up HPV test vs. none), vaccine product (brand_vacc: none, Cervarix, Gardasil 4, Gardasil 9), and vaccination program category (vacc_prog: none, childhood program in 7th grade, or catch-up program for women born 1991–1996 vaccinated during 2016–2019). Vaccination timing relative to LEEP was categorized as vaccination before the year of LEEP (pre_01), vaccination in the calendar year of LEEP (vacc_kon), or unvaccinated. Women whose first recorded HPV vaccine dose occurred after the first follow-up HPV test were classified as unvaccinated for the main analyses.

### 2.7. Statistical Analysis

Women were the unit of analysis. Categorical variables were summarized as counts and percentages. Associations between vaccination status (and vaccine type) and follow-up HPV outcomes were evaluated using Pearson’s χ^2^ tests (Fisher’s exact test where appropriate). Effect measures included absolute risk difference (percentage-point difference) with 95% confidence intervals.

To account for potential confounding, adjusted risk ratios (aRR) were estimated using modified Poisson regression with log link and robust variance. The main adjusted model used follow-up HPV positivity (yes/no) as the outcome and included vaccination in the calendar year of LEEP (yes/no), age at LEEP, year of LEEP, and pre-treatment HPV indicators (HPV16, HPV18, and pooled other HPV types) as covariates. This adjusted analysis was restricted to women without HPV vaccination prior to the year of LEEP and with a valid follow-up HPV result; women with missing covariate information were excluded. Two-sided *p*-values < 0.05 were considered statistically significant. Analyses were performed using IBM SPSS Statistics (v29).

### 2.8. Ethics, Data Protection, and Confidentiality

The follow-up protocol post-conization was appraised by the Regional Committee for Medical and Health Research Ethics (REK Nord) as a component of health service quality assurance (2015/2479/REK nord). The study was conducted within this quality-assurance framework at UNN using de-identified datasets. In accordance with Norwegian regulations for quality-assurance studies based on anonymized registry data, individual informed consent is not required. Data handling followed strict confidentiality procedures, and only anonymized analysis files and frequency tables were used for statistical analyses and manuscript preparation.

## 3. Results

In 2022–2024, LEEP specimens from 1052 women residing in Troms and Finnmark were processed at the Department of Clinical Pathology, University Hospital of North Norway (UNN). The mean age at LEEP was 39.8 years (SE 0.38; SD 12.3; range 22–79). All included women had a post-treatment follow-up HPV test registered in SymPathy and HPV vaccination status available in the national immunization register (SYSVAK), constituting the final study population. Descriptive distributions of follow-up HPV results and vaccination status by age group are shown in Table 1.

Overall, 329/1052 women (31.3%) were HPV-positive at the first follow-up after LEEP (Table 1). HPV positivity by treatment year was 33.4% (135/404) in 2022, 29.3% (115/393) in 2023, and 31.0% (79/255) in 2024. HPV positivity increased with age, from 25.5% (62/243) in women aged 20–29 years and 24.0% (89/371) in those aged 30–39 years, to 33.3% (68/204) in women aged 40–49 years and 37.7% (52/138) in those aged 50–59 years. The highest HPV positivity was observed among women aged 60–69 years (56.8%, 42/74) and 70–79 years (72.8%, 16/22).

Overall, HPV16 was detected in 81 women (7.7%) and HPV18 in 28 women (2.7%). The proportion with HPV16 also increased with age, from 5.3% (13/243) in women aged 20–29 years and 6.7% (25/371) in those aged 30–39 years, to 12.3% (17/138) in women aged 50–59 years, 13.5% (10/74) in those aged 60–69 years, and 27.3% (6/22) in women aged 70–79 years. HPV18 ranged from 1.4% (1/74) in women aged 60–69 years to 4.5% (1/22) in women aged 70–79 years.

Vaccination coverage decreased markedly with age. Overall, 522 women (49.6%) were vaccinated, with the highest coverage in women aged 20–29 years (79.4%, 193/243) and 30–39 years (53.6%, 199/371), declining to 34.8% (71/204) in women aged 40–49 years, 34.1% (47/138) in women aged 50–59 years, 14.9% (11/74) in women aged 60–69 years, and 4.5% (1/22) in women aged 70–79 years (Table 1).

Vaccine product distribution varied across age groups in a pattern consistent with the Norwegian vaccination programs and calendar-time effects (Table 2). Overall, Gardasil 9 was the most frequently recorded vaccine (330/1052), followed by Cervarix (174/1052) and Gardasil 4 (18/1052). Cervarix was most common in the youngest age group (20–29 years: 113/243, 46.5%) and decreased in 30–39 years (60/371, 16.2%), with minimal use in women aged ≥ 40 years (≤0.5%). Gardasil 9 predominated in women aged 30–59 years (30–39: 35.8%; 40–49: 33.8%; 50–59: 34.1%), while its use was lower in women aged 60–69 years (14.9%) and 70–79 years (4.5%). Gardasil 4 was uncommon across age groups (≤4.5%).

Vaccinated women had a lower proportion of positive follow-up HPV tests than unvaccinated women (24.7% [129/522] vs. 37.7% [200/530]; absolute risk reduction 13.0 percentage points [95% CI 7.5–18.6]; RR 0.655 [95% CI 0.544–0.788]; OR 0.542 [95% CI 0.415–0.706]; Pearson χ^2^ = 20.75, *p* = 5.2 × 10^−6^) (Table 3). Detection of HPV16 was also lower among vaccinated women (5.9% [31/522] vs. 9.4% [50/530]; Pearson χ^2^ = 4.52, *p* = 0.0335), while HPV18 detection did not differ between groups (2.9% [15/522] vs. 2.5% [13/530]; Pearson χ^2^ = 0.18, *p* = 0.671). For HPV types other than 16/18, positivity was 18.0% (94/522) in vaccinated women versus 28.7% (152/530) in unvaccinated women (Pearson χ^2^ = 16.72, *p* = 4.3 × 10^−5^).

In multivariable analysis restricted to women without vaccination prior to the year of LEEP and with a valid follow-up HPV result (*n* = 740), modified Poisson regression with robust variance showed that vaccination in the year of LEEP (vacc_kon) was associated with a lower risk of follow-up HPV positivity (aRR 0.592, 95% CI 0.444–0.789; *p* = 0.000348) after adjustment for age at LEEP, year of LEEP, and pre-treatment HPV indicators (HPV16, HPV18, and other HPV types). Increasing age was independently associated with higher risk of follow-up HPV positivity (aRR per year 1.021, 95% CI 1.013–1.029; *p* = 9.84 × 10^−8^), whereas year of LEEP was not (aRR 0.946, 95% CI 0.832–1.075; *p* = 0.394). Pre-treatment HPV16 and other HPV types were associated with higher risk of follow-up HPV positivity (aRR 1.569, 95% CI 1.222–2.016; *p* = 0.000423, and aRR 1.591, 95% CI 1.207–2.098; *p* = 0.000998, respectively), while HPV18 was not (aRR 1.306, 95% CI 0.925–1.842; *p* = 0.129) (Table 4).

Follow-up HPV positivity differed by vaccine type (Pearson χ^2^ = 25.89, df = 3, *p* < 0.001) (Table 5). The lowest HPV positivity was observed among women who had received Cervarix (18.4%, 32/174), followed by Gardasil 4 (22.2%, 4/18) and Gardasil 9 (28.2%, 93/330), while the highest positivity was seen in unvaccinated women (37.7%, 200/530).

Detection of HPV 16 varied markedly across vaccine types (Pearson χ^2^ = 19.59, df = 3, *p* < 0.001), with no HPV 16 detected among women vaccinated with Cervarix (0/174) or Gardasil 4 (0/18), whereas HPV 16 was detected in 9.4% of both Gardasil 9–vaccinated women (31/330) and unvaccinated women (50/530). In contrast, HPV 18 detection did not differ by vaccine type (Pearson χ^2^ = 2.38, df = 3, *p* = 0.497).

For HPV types other than 16/18, positivity also differed by vaccine type (Pearson χ^2^ = 17.11, df = 3, *p* = 0.001), being lowest for Cervarix (16.7%, 29/174) and Gardasil 9 (18.5%, 61/330), intermediate for Gardasil 4 (22.2%, 4/18), and highest among unvaccinated women (28.7%, 152/530).

Follow-up HPV positivity differed by vaccination timing (Pearson χ^2^, *p* < 0.001) (Table 6). The lowest HPV positivity was observed among women vaccinated in the calendar year of LEEP (20.5%, 44/215), followed by women vaccinated before LEEP (27.7%, 85/307), while the highest positivity was seen among unvaccinated women (37.7%, 200/530). Detection of HPV16 also differed by vaccination timing (Pearson χ^2^ = 6.48, df = 2, *p* = 0.039), with HPV16 detected in 4.6% (14/307) of women vaccinated before LEEP, 7.9% (17/215) of women vaccinated in the year of LEEP, and 9.4% (50/530) of unvaccinated women. In contrast, HPV18 detection did not differ across timing categories (Pearson χ^2^ = 0.38, df = 2, *p* = 0.825). For HPV types other than 16/18, positivity differed by vaccination timing (Pearson χ^2^, *p* < 0.001), being lowest among women vaccinated in the year of LEEP (11.6%, 25/215), intermediate among women vaccinated before LEEP (22.5%, 69/307), and highest among unvaccinated women (28.7%, 152/530). Women vaccinated after the follow-up HPV test were classified as unvaccinated for these analyses.

## 4. Discussion

In this population-based retrospective cohort of 1052 women treated by LEEP in Northern Norway (Troms and Finnmark) during 2022–2024, HPV vaccination prior to the first post-treatment follow-up was associated with a substantially lower probability of HPV positivity at the first registered follow-up HPV test. The association was evident for overall HPV positivity and for pooled HPV types other than 16/18, while no clear association was observed for HPV18. When stratified by vaccination timing, the lowest HPV positivity was observed among women vaccinated in the calendar year of LEEP, followed by women vaccinated before LEEP, with the highest positivity among unvaccinated women. In multivariable analyses restricted to women without vaccination prior to the year of LEEP, modified Poisson regression with robust variance showed that vaccination in the year of LEEP remained associated with a lower risk of follow-up HPV positivity (aRR 0.592, 95% CI 0.444–0.789) after adjustment for age, year of LEEP, and pre-treatment HPV indicators. Taken together, these findings support the hypothesis that vaccination around the time of excisional treatment may reduce early post-treatment HPV detection, a clinically relevant surrogate that is closely linked to subsequent risk of residual or recurrent disease.

### 4.1. Comparison with Previous UNN Data and External Evidence

Our findings are consistent with our previous regional cohort from 2022, where post-conization vaccination among women not vaccinated before treatment was associated with a higher probability of a negative HPV test at ~6 months (absolute risk reduction 12.1 percentage points; *p* = 0.039) [14]. Importantly, the present study extends this observation in three key ways: (i) by increasing sample size substantially, (ii) by including a broader age range (20–79 years), and (iii) by including three consecutive treatment years (2022–2024), thereby reducing the likelihood that the prior result reflected a chance finding or calendar-year–specific patterns.

The direction and magnitude of the association align with the broader literature suggesting a reduced risk of recurrent cervical dysplasia among vaccinated women after excisional treatment, although estimates vary and study quality is heterogeneous [4,5,6,10]. Systematic reviews and meta-analyses have generally reported risk reductions for CIN2+ recurrence after vaccination given at or around treatment, with the strongest evidence often observed for outcomes related to HPV16/18. At the same time, these reviews emphasize limitations such as reliance on observational data, differences in follow-up schedules, and incomplete control for confounding factors such as age and smoking [4,5,16,17].

Large observational studies have also contributed to the evidence base, including data suggesting reduced CIN2+ recurrence after post-excision vaccination, sometimes with attention to timing and margin status. Recent commentary has highlighted a persistent discrepancy between consistently “positive” pooled statistical estimates and uncertainty about biological mechanisms and residual bias in non-randomized studies, reinforcing the need for carefully designed studies with richer clinical covariate data [5,10,18,19].

### 4.2. Potential Explanations and Biological Plausibility

HPV vaccines are prophylactic and are not expected to eradicate an established infection on their own. The observed association with lower HPV positivity after LEEP may therefore reflect several non-mutually exclusive mechanisms:Prevention of new or repeated infection after treatment, including reinfection from partners or re-exposure, which could be particularly relevant in the months following excision when the transformation zone is healing.Reduction in reactivation and short-term viral spread to adjacent epithelium that remains after excision, mediated by vaccine-induced neutralizing antibodies.Immune microenvironment effects triggered by surgery and subsequent inflammatory signaling, which might theoretically enhance vaccine responsiveness or local immune control in the immediate post-operative period.

The lack of an apparent association for HPV18 in our data may be due to limited statistical power (given the low absolute number of HPV18-positive follow-up tests), different baseline genotype distributions, or differential natural history by genotype. It could also reflect that our HPV outcome categories are constrained by the available post-treatment test reporting (HPV16, HPV18, and pooled “other”), limiting fine-grained interpretation.

### 4.3. Age, Vaccine Uptake, and Vaccine Type Patterns

In our cohort, both HPV positivity after treatment and vaccine uptake varied markedly by age. Younger women had higher vaccine coverage, consistent with national vaccination program eligibility, whereas older women were less frequently vaccinated. These age gradients are important because age is associated with HPV persistence and with the probability of HPV detection after treatment and could confound vaccine–outcome associations if not fully controlled.

We also observed substantial differences in vaccine type distribution by age group, reflecting the Norwegian program history (routine school-based vaccination with quadrivalent vaccine in birth cohorts covered by the childhood program, and catch-up vaccination for selected cohorts) and the fact that many women receiving non-program vaccination (notably 9-valent vaccine) likely did so privately around the time of treatment. Because vaccine type is strongly correlated with age and calendar period, comparisons of HPV outcomes by vaccine product should be interpreted with caution and not taken as head-to-head effectiveness estimates. In particular, the apparent low HPV16 detection among women vaccinated with bivalent or quadrivalent vaccine may reflect cohort effects (earlier vaccination, different baseline risk, and different age structure), rather than a direct post-treatment product-specific effect.

### 4.4. Clinical Implications and Alignment with Guidance

Persistent HPV detection after treatment is clinically meaningful because it is associated with increased risk of residual or recurrent disease and is widely used in post-treatment surveillance. Our findings suggest that vaccination in connection with LEEP may reduce early HPV positivity, which could plausibly translate into fewer women requiring intensified surveillance and fewer repeat procedures—outcomes of particular importance given the reproductive and obstetric morbidity associated with repeat excisional treatments.

Professional guidance in some settings supports offering vaccination after treatment for CIN2+, often emphasizing shared decision-making and acknowledging remaining uncertainty in the evidence base [9]. Our results add real-world, registry-based evidence from a defined population with near-complete capture of vaccination status through SYSVAK, supporting the feasibility of monitoring post-treatment vaccination strategies at the population level.

### 4.5. Strengths

Key strengths of this study include the large sample size, inclusion of multiple treatment years, broad age coverage, and a population-based design within a geographically defined healthcare catchment area. Vaccination status was obtained from a national immunization registry (SYSVAK), reducing recall bias and improving completeness compared with self-reported vaccination. Outcome ascertainment through the laboratory information system (SymPathy) reflects real-world follow-up practice and provides clinically relevant HPV test results after treatment. The consistency of the association with our prior regional cohort further strengthens confidence in the robustness of the overall findings.

### 4.6. Limitations

Several limitations should be considered when interpreting these findings. First, the retrospective observational design is vulnerable to confounding and selection bias. Women who choose vaccination (particularly privately funded vaccination around treatment) may differ systematically from unvaccinated women in ways that influence HPV outcomes, including health-seeking behavior, smoking, immunosuppression, sexual behavior, and adherence to follow-up. Second, although we added an adjusted analysis, covariate availability was limited in these registry data; key clinical and behavioral factors such as lesion grade at treatment, margin status, smoking, immunosuppression, sexual behavior, and other risk markers were not available, and baseline HPV information was limited to HPV16, HPV18, and pooled “other” categories. Consequently, residual confounding cannot be excluded, and the adjusted estimates should be interpreted as associations rather than causal effects. These are well-recognized limitations in the broader literature on adjuvant vaccination [20,21,22]. Third, follow-up timing may not be fully standardized across patients, and HPV test results may represent a mix of persistent infection, new infection, or reactivation—entities that are difficult to distinguish in routine care datasets. Fourth, our outcome was HPV positivity at the first recorded follow-up (test-of-cure) rather than histologically confirmed recurrence, and HPV positivity is an imperfect surrogate that does not equate to relapse; the relationship between early HPV positivity and long-term CIN2+ recurrence may vary by genotype, age, and clinical context. The relatively low frequency of CIN2+ recurrence after LEEP and the limited follow-up time in this 2022–2024 cohort precluded robust analysis of long-term histologic endpoints. Finally, HPV genotype resolution in our setting was limited by the cobas assay to HPV16, HPV18, and a pooled ‘other’ category; therefore, we could not identify individual genotypes within the pool and could not distinguish vaccine-covered types (e.g., HPV31/33/45/52/58) from non-vaccine types. This constrains inference about genotype-specific effects beyond HPV16/18 and limits conclusions regarding whether reductions in the pooled ‘other’ category reflect changes in vaccine-type infections or shifts toward non-vaccine types.

### 4.7. Future Directions

Future studies should aim to (i) incorporate richer clinical data (margin status, lesion grade, baseline genotype, immunosuppression, smoking), (ii) standardize follow-up intervals or use time-to-event methods, and (iii) distinguish women vaccinated long before treatment from those vaccinated peritreatment, including dose number and precise timing. Given ongoing debate about biological mechanisms and the risk of residual bias in observational studies, studies that combine robust registry linkage with predefined analytic plans—and, where feasible, randomized designs—remain important to clarify causal effects and to estimate longer-term benefits and cost-effectiveness.

## 5. Conclusions

In a large, population-based cohort of women treated by LEEP in Northern Norway during 2022–2024, HPV vaccination prior to the first follow-up was associated with a markedly lower likelihood of HPV positivity at the first post-treatment follow-up. This real-world evidence supports vaccination as a potentially useful strategy to reduce early post-treatment HPV detection (test-of-cure) after excisional treatment and may reduce the need for intensified surveillance in women who remain HPV-positive. However, HPV positivity at first follow-up is a surrogate endpoint and does not equate to residual or recurrent CIN2+; studies with longer follow-up and histologic recurrence outcomes are needed to determine whether the observed reduction in early HPV detection translates into improved long-term clinical outcomes.

## Figures and Tables

**Figure 1 vaccines-14-00044-f001:**
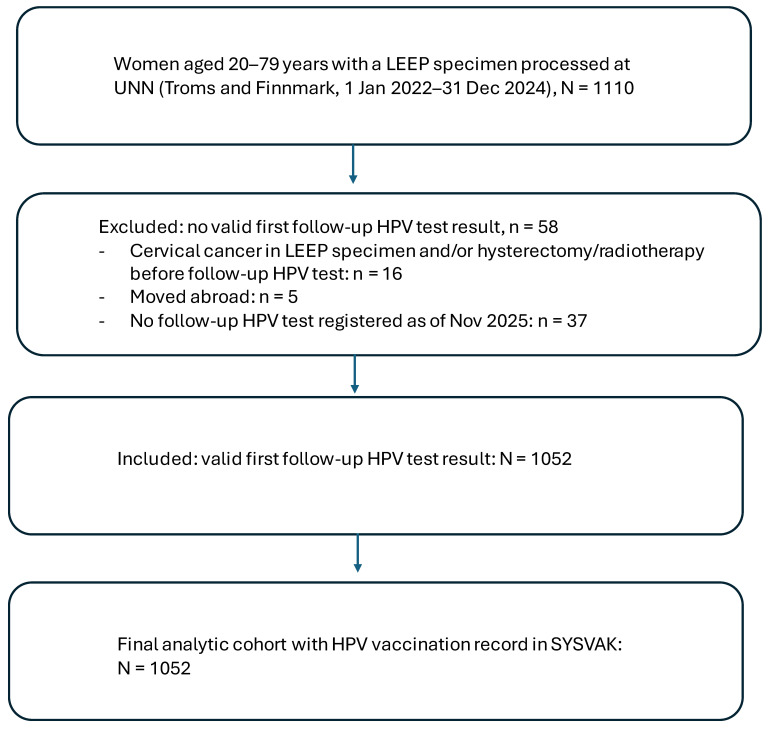
Flow diagram of study population.

**Table 1 vaccines-14-00044-t001:** HPV test positivity at first follow-up after LEEP, HPV16/HPV18, and vaccination status by age group.

Age Group	*N*	HPV Pos, *n* (%)	HPV16, *n* (%)	HPV18, *n* (%)	Vaccinated, *n* (%)
20–29 y	243	62 (25.5)	13 (5.3)	4 (1.6)	193 (79.4)
30–39 y	371	89 (24.0)	25 (6.7)	10 (2.7)	199 (53.6)
40–49 y	204	68 (33.3)	10 (4.9)	8 (3.9)	71 (34.8)
50–59 y	138	52 (37.7)	17 (12.3)	4 (2.9)	47 (34.1)
60–69 y	74	42 (56.8)	10 (13.5)	1 (1.4)	11 (14.9)
70–79 y	22	16 (72.8)	6 (27.3)	1 (4.5)	1 (4.5)
Total	1052	329 (31.3)	81 (7.7)	28 (2.7)	522 (49.6)

**Table 2 vaccines-14-00044-t002:** HPV vaccine type by age group.

Age Group	*N*	Cervarix, *n* (%)	Gardasil 4, *n* (%)	Gardasil 9, *n* (%)	Unvaccinated, *n* (%)
20–29 y	243	113 (46.5)	11 (4.5)	69 (28.4)	50 (20.6)
30–39 y	371	60 (16.2)	6 (1.6)	133 (35.8)	172 (46.4)
40–49 y	204	1 (0.5)	1 (0.5)	69 (33.8)	133 (65.2)
50–59 y	138	0 (0.0)	0 (0.0)	47 (34.1)	91 (65.9)
60–69 y	74	0 (0.0)	0 (0.0)	11 (14.9)	63 (85.1)
70–79 y	22	0 (0.0)	0 (0.0)	1 (4.5)	21 (95.5)
Total	1052	174 (16.5)	18 (1.7)	330 (31.4)	530 (50.4)

**Table 3 vaccines-14-00044-t003:** HPV vaccination status and HPV test results at first follow-up after LEEP.

HPV Vaccine	Total, *n*	HPV Pos, *n* (%)	HPV16, *n* (%)	HPV18, *n* (%)	Other HPV, *n* (%)
Yes	522	129 (24.7)	31 (5.9)	15 (2.9)	94 (18.0)
No	530	200 (37.7)	50 (9.4)	13 (2.5)	152 (28.7)
Total	1052	329 (31.3)	81 (7.7)	28 (2.7)	246 (23.4)

**Table 4 vaccines-14-00044-t004:** Adjusted risk ratios (aRR) for follow-up HPV positivity after LEEP (modified Poisson regression with robust variance). Population: women without vaccination prior to the year of LEEP and with a valid follow-up HPV result (*n* = 740). Outcome: follow-up HPV positivity (yes vs. no).

Predictor	aRR	95% CI	*p*-Value
Vaccinated in the year of LEEP	0.592	0.444–0.789	0.000348
Age at LEEP (per year)	1.021	1.013–1.029	9.84 × 10^−8^
Year of LEEP (per year)	0.946	0.832–1.075	0.394
Pre-treatment HPV16	1.569	1.222–2.016	0.000423
Pre-treatment HPV18	1.306	0.925–1.842	0.129
Pre-treatment other HPV types	1.591	1.207–2.098	0.000998

Model adjusted for all variables shown; odds ratios from logistic regression.

**Table 5 vaccines-14-00044-t005:** HPV vaccine type and HPV test results at first follow-up after LEEP.

Vaccine Type	Total, *n*	HPV Pos, *n* (%)	HPV16, *n* (%)	HPV18, *n* (%)	Other HPV, *n* (%)
Cervarix	174	32 (18.4)	0 (0.0)	3 (1.7)	29 (16.7)
Gardasil 4	18	4 (22.2)	0 (0.0)	0 (0.0)	4 (22.2)
Gardasil 9	330	93 (28.2)	31 (9.4)	12 (3.6)	61 (18.5)
None	530	200 (37.7)	50 (9.4)	13 (2.5)	152 (28.7)
Total	1052	329 (31.3)	81 (7.7)	28 (2.7)	246 (23.4)

**Table 6 vaccines-14-00044-t006:** HPV test results at first follow-up after LEEP, stratified by vaccination timing.

Vaccination Timing Category	Total, *n*	HPV Pos, *n* (%)	HPV16, *n* (%)	HPV18, *n* (%)	Other HPV, *n* (%)
Vaccinated before LEEP	307	85 (27.7)	14 (4.6)	8 (2.6)	69 (22.5)
Vaccinated in the calendar year of LEEP	215	44 (20.5)	17 (7.9)	7 (3.3)	25 (11.6)
Unvaccinated	530	200 (37.7)	50 (9.4)	13 (2.5)	152 (28.7)
Total	1052	329 (31.3)	81 (7.7)	28 (2.7)	246 (23.4)

Percentages are within vaccination timing category. Pearson χ^2^
*p*-values across the three timing categories: HPV positivity *p* < 0.001; HPV16 *p* = 0.039; HPV18 *p* = 0.825; other HPV *p* < 0.001. Women vaccinated after the follow-up HPV test were classified as unvaccinated for these analyses.

## Data Availability

Restrictions apply to the availability of these data. The datasets analyzed in this study contain individual-level health information from the SymPathy laboratory information system (University Hospital of North Norway) and the Norwegian Immunization Registry (SYSVAK). Due to privacy and legal restrictions, these data cannot be made publicly available. Aggregated data are included in the article. De-identified data underlying the results may be made available from the corresponding author upon reasonable request and subject to applicable approvals and data access permissions.

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
