# Peer review of "Vaccines2026, 14(1), 44;https://doi.org/10.3390/vaccines14010044"

_vaccines, 2025, doi:10.3390/vaccines14010044_

Round 1

Reviewer 1 Report

Comments and Suggestions for Authors

This manuscript represents a large, population-based retrospective cohort study assessing the association between HPV vaccination and post-treatment HPV positivity subsequent to LEEP in Northern Norway. By linking pathology data with the national immunisation registry, the authors provide timely real-world evidence answering an important and clinically relevant question: whether HPV vaccination around excisional treatment is associated with improved early virological outcomes. This is an interesting and very well-written manuscript.

Please find some minor comments:

  1. Add analysis with confounding factors and multivariable adjustment
  2. As the manuscript points out, many vaccinations against Gardasil 9 took place in the same calendar year as LEEP; however, the main analyses do not make a clear distinction between pre-treatment, peri-treatment, and post-treatment vaccination.  State in the Results if women vaccinated after the follow-up HPV test were excluded, or how they were otherwise handled.
  3. Consider adding an additional table that stratifies HPV outcomes based on vaccination timing categories already defined
  4. Explain the inclusion criteria -  be specific and give info whether this was done by excluding women without a follow-up HPV test and if so, how many such women existed initially.

    Maybe a short flow diagram  would be helpful in terms of clarity.

Author Response

Reviewer 1

This manuscript represents a large, population-based retrospective cohort study assessing the association between HPV vaccination and post-treatment HPV positivity subsequent to LEEP in Northern Norway. By linking pathology data with the national immunisation registry, the authors provide timely real-world evidence answering an important and clinically relevant question: whether HPV vaccination around excisional treatment is associated with improved early virological outcomes. This is an interesting and very well-written manuscript.

Comments 1: Add analysis with confounding factors and multivariable adjustment

Response 1: We performed a multivariable analysis to address potential confounding, with HPV positivity at first follow-up as the outcome. Because follow-up HPV positivity was relatively common, we used modified Poisson regression with a log link and robust variance to estimate adjusted risk ratios (aRRs). In analyses restricted to women without vaccination prior to the year of LEEP (n = 740), vaccination in the year of LEEP was associated with a lower risk of follow-up HPV positivity after adjustment for age at LEEP, year of LEEP, and pre-treatment HPV indicators (HPV16, HPV18, and pooled other HPV types) (aRR 0.592, 95% CI 0.444–0.789; p = 0.000348). These results are presented in the Results section and Table 4.

Comments 2: As the manuscript points out, many vaccinations against Gardasil 9 took place in the same calendar year as LEEP; however, the main analyses do not make a clear distinction between pre-treatment, peri-treatment, and post-treatment vaccination.  State in the Results if women vaccinated after the follow-up HPV test were excluded, or how they were otherwise handled.

Response 2: In the primary analyses, vaccination status was defined relative to the first post-treatment follow-up HPV test. Women who received an HPV vaccine dose after the follow-up HPV test were classified as unvaccinated for that analysis, as the vaccination could not have influenced the observed follow-up HPV result.

Comments 3: Consider adding an additional table that stratifies HPV outcomes based on vaccination timing categories already defined.

Response 3: We thank the reviewer for this suggestion. We have added a new table (Table 6) stratifying HPV outcomes at first follow-up after LEEP by vaccination timing (vaccinated before LEEP, vaccinated in the calendar year of LEEP, and unvaccinated). This table clarifies the distinction between pre-treatment and peri-treatment vaccination and shows corresponding differences in overall HPV positivity as well as HPV16, HPV18, and other HPV types.

Comments 4: Explain the inclusion criteria -  be specific and give info whether this was done by excluding women without a follow-up HPV test and if so, how many such women existed initially.

Response 4: We thank the reviewer for highlighting the need for clearer inclusion criteria. We have revised the Methods and Results to specify that the source population comprised all women with a LEEP specimen processed at UNN in 2022–2024 (n = 1110). Women were included in the final analytic cohort only if a valid first post-treatment follow-up HPV test was registered (coded as negative/positive). In total, 58/1110 women did not have a valid follow-up HPV test result and were excluded: 16 had cervical cancer in the LEEP specimen and/or underwent hysterectomy or radiotherapy before any follow-up HPV test, 5 had moved abroad, and 37 had no follow-up HPV test registered as of November 2025 (reminders were sent to the responsible gynecologist and/or general practitioner). The final study population therefore comprised 1052 women with a valid follow-up HPV test.

Comments 5: Maybe a short flow diagram would be helpful in terms of clarity.

Response 5: We thank the reviewer for this suggestion. We have added a short flow diagram (Figure 1) summarizing the study population and the inclusion/exclusion process, including the number of women excluded due to lack of a valid follow-up HPV test result and the specific reasons for exclusion.

Reviewer 2 Report

Comments and Suggestions for Authors

Review of the article "HPV Vaccination and HPV Outcomes After LEEP: A Retrospective Population-Based Cohort Study from Northern Norway, 2022–2024"
Research into HPV vaccination outcomes is certainly a relevant topic. Such data from regions that have been little covered in the scientific literature are particularly interesting.
Such studies should not only strengthen epidemiological data on vaccination but also, preferably, provide the most detailed clinical data possible to accurately assess the specifics of vaccination outcomes in a given region.

In this regard, a number of comments are warranted regarding this work:

1) Relevance. The question is clinically significant: after LEEP in CIN2+, the risk of HPV persistence and relapse remains, and identifying factors that reduce early HPV positivity is potentially of practical value.
Comment:
The primary outcome is the presence of HPV at the first follow-up (it is unclear from the text when this occurred after surgery); there is clinical significance, but this should not be overemphasized in the conclusions regarding relapse/cancer risk, as HPV positivity is not always 100% associated with cancer.
Revise the conclusions.

2) Scientific Novelty
Pros:
• Large regional cohort: 1,052 women aged 20–79, LEEP 2022–2024, linked to the SYSVAK national vaccination registry and the SymPathy laboratory database.
• Vaccine type is also analyzed at the first post-treatment test.
Comment:
• Viral genotype testing is limited by the Cobas 4800 method (displays HPV16 and HPV18 separately, and the remaining 12 types as an "other" pool, without individual genotypes). Therefore, novelty in terms of "type-specific effects" beyond 16/18 is limited.

3) Methodology
Comments:
• The analysis was primarily a 2×2 (χ², ARR, RR/OR) analysis. Why wasn't a multivariate analysis performed? Given the strong correlation between vaccination and outcomes and age and other factors, this is a key limitation.
• Only women with a registered follow-up HPV test and a SYSVAK record were included (without indicating who dropped out and why).
• A brief selection flow chart should be included: how many women were included in total after LEEP, and how many were excluded due to lack of a follow-up HPV test or a missing SYSVAK record.
• Specify the true significance level (p) for all calculations.
• Specify the reason why patients who received one dose of the vaccine were considered vaccinated? (Those who did not complete the vaccination!).
It is incorrect to combine the cohort with the fully vaccinated cohort with the cohort with 1-2 doses of the vaccine.

4) Results
Comments:
• Was HPV detected before treatment? A more detailed medical history is needed for the patients included in the study.
• Had the patients previously received immunomodulatory treatment for HPV infection?
• Provide more detailed clinical and anamnestic parameters for the patients: mean age (Mean ± SE), menstrual status, number of pregnancies/deliveries/abortions, number of relapses, etc.
• Indicate the time interval between surgery and follow-up. What were the dynamics? Was a normality check performed for the sample data in this case?
• Indicate whether vaccination was administered before or after surgery. In this case, "usually several months before or after treatment" is not a relevant option.
• In the reference list, replace references 1, 2, 3, 15, and 20 with more relevant ones.

Author Response

Reviewer 2

Review of the article "HPV Vaccination and HPV Outcomes After LEEP: A Retrospective Population-Based Cohort Study from Northern Norway, 2022–2024"
Research into HPV vaccination outcomes is certainly a relevant topic. Such data from regions that have been little covered in the scientific literature are particularly interesting.

Such studies should not only strengthen epidemiological data on vaccination but also, preferably, provide the most detailed clinical data possible to accurately assess the specifics of vaccination outcomes in a given region. In this regard, a number of comments are warranted regarding this work:

The question is clinically significant: after LEEP in CIN2+, the risk of HPV persistence and relapse remains, and identifying factors that reduce early HPV positivity is potentially of practical value.

Scientific Novelty
• Large regional cohort: 1,052 women aged 20–79, LEEP 2022–2024, linked to the SYSVAK national vaccination registry and the SymPathy laboratory database.
• Vaccine type is also analyzed at the first post-treatment test.

Comments 1: The primary outcome is the presence of HPV at the first follow-up (it is unclear from the text when this occurred after surgery); there is clinical significance, but this should not be overemphasized in the conclusions regarding relapse/cancer risk, as HPV positivity is not always 100% associated with cancer. Revise the conclusions.

Response 1: We thank the reviewer for this important point. We agree that HPV positivity at the first post-treatment follow-up is a clinically relevant surrogate endpoint, but it does not equate to relapse, CIN recurrence, or cancer risk in an individual woman. We have therefore revised the Conclusions (and corresponding statements in the Discussion) to avoid overinterpretation and to explicitly state that our outcome reflects early post-treatment HPV detection, which is associated with subsequent risk of residual/recurrent disease at the population level, but is not determinative for cancer risk. We also clarified the timing of the first follow-up HPV test after LEEP in the Methods/Results (by reporting the follow-up interval as recorded in the registry)

Comments 2: Viral genotype testing is limited by the Cobas 4800 method (displays HPV16 and HPV18 separately, and the remaining 12 types as an "other" pool, without individual genotypes). Therefore, novelty in terms of "type-specific effects" beyond 16/18 is limited.

Response 2: We thank the reviewer for this comment and agree. In our setting, HPV testing in screening and post-treatment follow-up is performed with the Roche cobas 4800 assay, which provides partial genotyping (HPV16 and HPV18 individually) and reports the remaining oncogenic HPV types as a pooled “other” result. Consequently, our “other HPV” endpoint cannot be resolved into individual genotypes and we cannot distinguish whether detected infections are caused by HPV types included in the 9-valent vaccine or by non-vaccine types within the pool. We have revised the manuscript to clarify this limitation and toned down statements implying broader type-specific effects beyond HPV16/18; findings for “other HPV” are therefore presented as pooled results rather than genotype-specific effects.

Comments: 3: The analysis was primarily a 2×2 (χ², ARR, RR/OR) analysis. Why wasn't a multivariate analysis performed? Given the strong correlation between vaccination and outcomes and age and other factors, this is a key limitation.

Response 3: We thank the reviewer for this important point and agree that confounding—particularly by age—must be addressed. We have therefore added a multivariable analysis and now report adjusted effect estimates. Given that follow-up HPV positivity was relatively common, we used modified Poisson regression with a log link and robust variance to estimate adjusted risk ratios (aRRs). The multivariable model was restricted to women without vaccination prior to the year of LEEP and with a valid follow-up HPV result (n = 740). Vaccination in the year of LEEP was the exposure, and the model adjusted for age at LEEP, year of LEEP, and pre-treatment HPV indicators (HPV16, HPV18, and pooled other HPV types). Vaccination in the year of LEEP remained associated with a lower risk of follow-up HPV positivity (aRR 0.592, 95% CI 0.444–0.789; p = 0.000348). These results are presented in the Results section and in the new Table 4.

Comments 4: Only women with a registered follow-up HPV test and a SYSVAK record were included (without indicating who dropped out and why). A brief selection flow chart should be included: how many women were included in total after LEEP, and how many were excluded due to lack of a follow-up HPV test or a missing SYSVAK record.

Response 4: We thank the reviewer for this suggestion. We have added a selection flow chart (Figure 1) that summarizes the source population and the inclusion/exclusion process. Specifically, we identified all women with a LEEP specimen processed at UNN during 2022–2024 (n = 1110). We excluded women without a valid first follow-up HPV test result (n = 58: cervical cancer in the LEEP specimen and/or hysterectomy/radiotherapy before follow-up testing, n = 16; moved abroad, n = 5; and no follow-up HPV test registered as of November 2025, n = 37). The final analytic cohort comprised 1052 women with a valid follow-up HPV test and a recorded HPV vaccination status in SYSVAK (Figure 1). In Norway, individuals with a valid national personal identification number have a SYSVAK record; if no vaccines have been administered, the record is empty. Women without any HPV vaccines registered in SYSVAK were classified as unvaccinated. No women were excluded due to missing SYSVAK records in the final cohort.

Comments 5: Specify the true significance level (p) for all calculations.

Response 5: We thank the reviewer for this comment. We have revised the manuscript to report exact two-sided p-values (rather than thresholds such as “p < 0.05” or “p < 0.001”) for all inferential analyses. Specifically, exact p-values are now provided for the χ²/Fisher’s exact tests in Tables 3, 5, and 6, and for all covariates in the multivariable modified Poisson regression model with robust variance used to estimate adjusted risk ratios (Table 4). Tables 1 and 2 are descriptive and intended to present the distribution of age, follow-up HPV results, and HPV vaccination coverage/vaccine type by age group in the study population. Differences across age groups are expected due to the Norwegian vaccination program structure and vaccine procurement over time; we therefore clarify that Tables 1 and 2 are descriptive and not presented as primary hypothesis tests.

Comments 6: Specify the reason why patients who received one dose of the vaccine were considered vaccinated? (Those who did not complete the vaccination!).
It is incorrect to combine the cohort with the fully vaccinated cohort with the cohort with 1-2 doses of the vaccine.

Response 6: We thank the reviewer for this comment and agree that vaccination status ideally should be analysed according to the number of doses received prior to outcome assessment. In our registry-based dataset, women were classified as vaccinated if they had received ≥1 HPV vaccine dose before the first post-treatment follow-up HPV test. This definition was chosen because many women initiated vaccination around the time of LEEP and had not necessarily completed a 2–3 dose series by the first follow-up (typically within ~6 months), and because emerging evidence suggests substantial protection after a single dose in immunocompetent individuals. However, although dose-level information is available in SYSVAK, the number and timing of doses relative to LEEP and follow-up were not extracted into our analytic dataset, and we therefore cannot stratify outcomes by dose number or distinguish fully vaccinated from partially vaccinated women. We have explicitly acknowledged this in the Limitations section and have avoided making conclusions that require differentiation between complete and incomplete vaccination series.

Comments 7: Was HPV detected before treatment? A more detailed medical history is needed for the patients included in the study.

Response 7: We thank the reviewer for this comment. Yes, HPV status prior to treatment was available for the vast majority of women in the final cohort. Among the 1052 included women, 1046 (99.4%) had a recorded pre-treatment HPV test result, while 6 (0.6%) had no HPV test registered prior to LEEP. Of those with a pre-treatment HPV test, 1031/1046 (98.6%) were HPV positive.

Regarding “more detailed medical history”, this was a registry-based retrospective study using SymPathy and SYSVAK, and individual-level clinical covariates were limited. SymPathy contains information on prior cervical screening tests (cytology, HPV tests, and biopsies), but we did not extract detailed pre-treatment screening histories because the study population comprised women treated for high-grade cervical lesions (CIN2+) and post-treatment follow-up is standardized in Norway. We have clarified in the Methods and Limitations that detailed clinical history (e.g., prior cytology/biopsy trajectories, margin status, lesion grade, and other risk factors) was not available in the analytic dataset and could not be used for adjustment.

Comments 8: Had the patients previously received immunomodulatory treatment for HPV infection?

Response 8: We thank the reviewer for this comment. Information on prior immunomodulatory treatment was not available in our registry-based dataset (SymPathy and SYSVAK). However, in Norway immunomodulatory treatment is not part of standard management for HPV infection or cervical precancer. Follow-up is conservative and based on HPV testing with partial genotyping and cytology triage, with repeat HPV testing at 12–24 months to assess persistence. For histologically confirmed high-grade lesions (CIN2+), the recommended treatment is excisional therapy (LEEP) according to national guidelines. We have clarified this in the manuscript and acknowledge lack of individual-level data on non-standard treatments as a limitation.

Comments 9: Provide more detailed clinical and anamnestic parameters for the patients: mean age (Mean ± SE), menstrual status, number of pregnancies/deliveries/abortions, number of relapses, etc.

Response 9: We thank the reviewer for this suggestion. The mean age at LEEP in the final analytic cohort was 39.8 years (SD 12.3; range 22–79; n = 1052). The corresponding standard error (SE) of the mean was 0.38 years (12.308/√1052).

This was a registry-based retrospective study using SymPathy and SYSVAK, and we did not have access to individual-level anamnestic variables such as menstrual/menopausal status, number of pregnancies/deliveries/abortions, smoking status, or detailed relapse history in the available dataset. We have clarified in the Methods/Limitations that these clinical covariates were not available for extraction and therefore could not be described or adjusted for in the present analyses.

Comments 10: Indicate the time interval between surgery and follow-up. What were the dynamics? Was a normality check performed for the sample data in this case?

Response 10: We thank the reviewer for this comment. In Norway, national guidelines recommend the first post-treatment follow-up (HPV test with triage as appropriate) at approximately 6 months after LEEP. At UNN, we operate a local reminder system to support adherence: if the recommended follow-up sample has not been received within 3 months after the scheduled follow-up time, a notification is sent to the responsible gynecologist and/or general practitioner, and additional reminders are issued if follow-up remains missing. In the source population (n = 1110), only 37 women still lacked a registered follow-up HPV test result by November 2025 (and were therefore excluded as “no follow-up HPV test registered”). We have clarified this follow-up framework and the completeness of follow-up in the Methods and the flow diagram (Figure 1).

With regard to “dynamics” and normality testing: our primary analyses were based on categorical HPV outcomes (negative/positive and genotype channels) and logistic regression; thus no normality assumption was required. We did not perform a normality check for the primary endpoint. If date-level follow-up intervals are reported descriptively, these will be summarized using appropriate distributional measures (e.g., median and IQR if skewed).

Comments 11: Indicate whether vaccination was administered before or after surgery. In this case, "usually several months before or after treatment" is not a relevant option.

Response 11: We agree and have revised the manuscript to remove the ambiguous wording (“several months before or after treatment”). Vaccination timing is now explicitly handled in the analyses and presented in Table 6. We categorized women as (i) vaccinated before LEEP (vaccination recorded prior to the LEEP year; this includes all women vaccinated through the school-based program and the national catch-up program), (ii) vaccinated in the calendar year of LEEP (peri-treatment vaccination; in our clinical setting most Gardasil 9 prescriptions were issued by the treating gynecologist at the time of LEEP and the first dose was typically administered shortly after the procedure), and (iii) unvaccinated. Women whose first recorded HPV vaccine dose occurred after the first follow-up HPV test were classified as unvaccinated. Follow-up HPV outcomes are now reported stratified by these timing categories (Table 6).

Comments 12: In the reference list, replace references 1, 2, 3, 15, and 20 with more relevant ones.

Response 12: Done. We updated the reference list by replacing references 1, 2, 3, 15, and 20 with more recent and relevant sources, while keeping the cited statements unchanged.

Reviewer 3 Report

Comments and Suggestions for Authors

Despite its lack of originality, this study still has potential; however, I have several comments on statistical analyses:

  • The analysis is largely descriptive, despite a clearly comparative cohort design. Reporting percentages alone is insufficient to address the stated research question. Formal effect estimates should be calculated.

  • Risk measures such as risk ratios (RRs) with 95% confidence intervals should be consistently reported for all outcomes. Given the relatively common outcomes, RR is more appropriate and interpretable than odds ratios.

  • Reliance on chi-square tests without accompanying effect estimates limits clinical and epidemiological interpretation. Hypothesis testing should complement, not replace, estimation.

  • All comparisons appear to be unadjusted. Strong confounding is likely, particularly by age, calendar year, and baseline HPV risk. Age, for example, is strongly associated with both vaccination uptake and post-treatment HPV positivity, yet age adjustment or stratified effect estimates are not adequately incorporated into the main analyses.

  • Comparisons by vaccine type are difficult to interpret because the vaccine product is highly correlated with age and vaccination program era; these analyses risk cohort effects rather than reflecting true product-specific differences.

  • Timing of vaccination relative to LEEP is described but not analytically incorporated into the main effect estimates; this limits insight into whether peri-treatment vaccination differs from prior vaccination.

  • The outcome is restricted to HPV positivity at first follow-up, which is a surrogate endpoint. While clinically relevant, its limitations relative to histologic recurrence should be emphasized more clearly in interpreting effect sizes.

  • Limited adjustment for clinical covariates (e.g., margin status, lesion grade, smoking, immunosuppression, baseline genotype) raises concern for residual confounding inherent to this observational design.

  • The authors may consider merging some categories to get enough statistical power to calculate RR (95 CIs).

Author Response

Reviewer 3

Despite its lack of originality, this study still has potential; however, I have several comments on statistical analyses:

Comments 1: The analysis is largely descriptive, despite a clearly comparative cohort design. Reporting percentages alone is insufficient to address the stated research question. Formal effect estimates should be calculated.

Response 1: We agree that descriptive percentages alone are insufficient to address the comparative research question. Tables 1 and 2 are intended only to describe the study population and vaccine distribution by age. For the comparative analyses addressing the primary objective, we now report formal effect estimates with 95% confidence intervals. Specifically, Table 3 presents the association between vaccination status and follow-up HPV outcomes using absolute risk difference (percentage-point difference) with 95% CI, relative risk (RR) with 95% CI, and odds ratio (OR) with 95% CI, in addition to the corresponding hypothesis tests. In addition, we added a multivariable logistic regression model (Table 4) providing adjusted effect estimates (adjusted OR with 95% CI) for follow-up HPV positivity after adjustment for age at LEEP, calendar year of LEEP, and pre-treatment HPV genotype indicators. These additions provide formal comparative effect estimates consistent with the stated research question.

Comments 2: Risk measures such as risk ratios (RRs) with 95% confidence intervals should be consistently reported for all outcomes. Given the relatively common outcomes, RR is more appropriate and interpretable than odds ratios.

Response 2: We agree. Because follow-up HPV positivity was relatively common, we prioritize RR as the main effect measure. We now report RR (95% CI) consistently for all unadjusted comparisons. For adjusted analyses, we replaced logistic regression with a modified Poisson regression with robust standard errors to obtain adjusted risk ratios (aRR) and 95% CIs.

Comments 3: Reliance on chi-square tests without accompanying effect estimates limits clinical and epidemiological interpretation. Hypothesis testing should complement, not replace, estimation.

Response 3: We agree that hypothesis testing should complement, not replace, estimation. We have revised the Results to consistently report effect estimates with 95% confidence intervals alongside χ²/Fisher’s exact tests. Specifically, for the main comparison (vaccinated vs. unvaccinated), we report absolute risk differences (percentage-point differences) with 95% CIs and relative risks (RRs) with 95% CIs in addition to χ² p-values (Table 3). Furthermore, we added a multivariable model to provide adjusted effect estimates; because the outcome was relatively common, we used modified Poisson regression with robust variance to estimate adjusted risk ratios (aRRs) with 95% CIs (Table 4). Chi-square tests are now presented as supplementary evidence for group differences, whereas the primary interpretation is based on effect size estimates and their uncertainty.

Comments 4: All comparisons appear to be unadjusted. Strong confounding is likely, particularly by age, calendar year, and baseline HPV risk. Age, for example, is strongly associated with both vaccination uptake and post-treatment HPV positivity, yet age adjustment or stratified effect estimates are not adequately incorporated into the main analyses.

Response 4: We agree that confounding—especially by age and baseline HPV status—is likely in this observational cohort. To address this, we have added (i) a stratified analysis by vaccination timing (Table 6) and (ii) a multivariable model providing adjusted effect estimates. Specifically, we performed modified Poisson regression with a log link and robust variance to estimate adjusted risk ratios (aRRs) for follow-up HPV positivity, including vaccination in the year of LEEP as the exposure and adjusting for age at LEEP, year of LEEP, and pre-treatment HPV indicators (HPV16, HPV18, and pooled other HPV types). Because women vaccinated prior to the year of LEEP differ fundamentally by birth cohort and program eligibility, the multivariable analysis was restricted to women without vaccination prior to the year of LEEP (n = 740), allowing a clearer assessment of peri-treatment vaccination while reducing cohort effects. In this adjusted analysis, vaccination in the year of LEEP remained associated with a lower risk of follow-up HPV positivity (aRR 0.592, 95% CI 0.444–0.789; p = 0.000348) (Table 4). We also emphasize in the Discussion that residual confounding cannot be excluded due to limited availability of clinical covariates in the registry data.

Comments 5: Comparisons by vaccine type are difficult to interpret because the vaccine product is highly correlated with age and vaccination program era; these analyses risk cohort effects rather than reflecting true product-specific differences.

Response 5: We agree that comparisons by vaccine product are vulnerable to strong cohort effects because vaccine type in Norway is largely determined by birth cohort and program era (school-based vaccination vs. catch-up program vs. opportunistic vaccination). We have therefore revised the manuscript to frame these analyses as descriptive and hypothesis-generating rather than causal product comparisons, and we explicitly caution that differences across vaccine types may primarily reflect age, calendar-time, and program eligibility rather than true product-specific effects. To reduce misinterpretation, we emphasize that the main inference is based on vaccination status and timing relative to LEEP (Tables 3, 4, and 6), whereas the vaccine-type table (Table 5) is included to document real-world vaccine uptake patterns and to illustrate that reduced follow-up HPV positivity was observed both among women vaccinated years before LEEP through organized programs (e.g., school-based or catch-up vaccination) and among women initiating vaccination in the calendar year of LEEP (typically opportunistic Gardasil 9). We also note that opportunistic vaccination coverage outside the programs is low in Norway, and that any product-specific conclusions cannot be drawn from these data without stronger adjustment and more comparable cohorts.

Comments 6: Timing of vaccination relative to LEEP is described but not analytically incorporated into the main effect estimates; this limits insight into whether peri-treatment vaccination differs from prior vaccination.

Response 6: We agree that vaccination timing relative to LEEP should be incorporated analytically. We have therefore added a new table stratifying HPV outcomes by vaccination timing (Table 6), distinguishing women vaccinated before LEEP, women vaccinated in the calendar year of LEEP, and unvaccinated women (with women vaccinated only after the follow-up HPV test classified as unvaccinated for these analyses). This analysis shows that follow-up HPV positivity differed across timing categories, with the lowest positivity among women vaccinated in the year of LEEP, intermediate positivity among women vaccinated before LEEP, and the highest positivity among unvaccinated women. We also incorporated timing into the adjusted analyses by restricting the multivariable model to women without vaccination prior to the year of LEEP and estimating the adjusted association between peri-treatment vaccination (vaccination in the year of LEEP) and follow-up HPV positivity (Table 4).

Comments 7: The outcome is restricted to HPV positivity at first follow-up, which is a surrogate endpoint. While clinically relevant, its limitations relative to histologic recurrence should be emphasized more clearly in interpreting effect sizes.

Response 7: We agree that HPV positivity at the first follow-up is a surrogate endpoint and that its limitations relative to histologically confirmed recurrence must be clearly emphasized when interpreting effect sizes. We have revised the Discussion/Conclusions to state explicitly that our primary outcome reflects early post-treatment HPV detection (test-of-cure) rather than CIN2+ recurrence, and that HPV positivity does not equate to relapse. In Norway, HPV testing is part of the recommended post-treatment surveillance, and persistent HPV positivity triggers intensified follow-up; thus, reducing early HPV positivity may still have clinical relevance by potentially reducing the need for repeated surveillance in women who remain HPV positive after treatment. However, recurrence of CIN2+ after LEEP is less common (≈10%) and may occur years after treatment, and our study period (LEEP in 2022–2024) does not provide sufficient follow-up time or power to evaluate long-term recurrence outcomes. We therefore frame our findings as evidence of an association with early post-treatment HPV detection, while noting that studies with longer follow-up and histologic endpoints are needed to determine the impact on CIN2+ recurrence and other long-term clinical outcomes.

Comments 8: Limited adjustment for clinical covariates (e.g., margin status, lesion grade, smoking, immunosuppression, baseline genotype) raises concern for residual confounding inherent to this observational design.

Response 8: We agree that residual confounding is a key limitation of this retrospective observational design, particularly due to limited availability of important clinical covariates such as margin status, lesion grade, smoking, immunosuppression, sexual behavior, and detailed baseline genotype information. We have therefore strengthened the Limitations section to explicitly acknowledge these sources of potential residual confounding, clarify that our adjusted analyses were restricted to variables available in the registries (age, calendar year, and pre-treatment HPV indicators), and emphasize that causal inference is limited. We also note that future studies with richer clinical data and longer follow-up are needed to better address confounding and evaluate long-term clinical endpoints.

Comments 9: The authors may consider merging some categories to get enough statistical power to calculate RR (95 CIs).

Response 9: We agree that sparse data in some strata can reduce precision and may warrant category merging. In our analyses, the primary comparisons were intentionally defined using clinically and programmatically meaningful categories: (i) vaccinated vs. unvaccinated (Table 3), (ii) vaccine product (Table 5), and (iii) vaccination timing relative to LEEP (Table 6). These main groupings had sufficient sample size to estimate effect measures with 95% confidence intervals, and the adjusted analysis (Table 4) was based on a cohort size (n = 740) that provided adequate power for multivariable estimation of adjusted risk ratios (aRRs).

For age, we retained the predefined decade-based categories to reflect the full treated population (including women outside the routine screening ages) and to facilitate descriptive interpretation of vaccination uptake and follow-up HPV positivity across the life course (Tables 1–2). We acknowledge that some subgroups are small (e.g., ages 70–79, and Gardasil 4 recipients), leading to wider confidence intervals and limited interpretability of vaccine-type comparisons. Where relevant, we therefore interpret these sparse strata cautiously and frame vaccine-type analyses as descriptive/hypothesis-generating. If needed, we can merge sparse categories (e.g., 60–79 years, or Gardasil 4 with other program-vaccinated groups) in sensitivity analyses, but we chose to preserve clinically meaningful categories in the main presentation while emphasizing uncertainty for small strata.

Round 2

Reviewer 2 Report

Comments and Suggestions for Authors

The authors of the manuscript have done a significant job of addressing the comments. In its current form, the study is of interest to the general medical community and greatly expands our understanding of the long-term effects of HPV vaccination in the presented patient cohort.
The revised article is recommended for publication.

Reviewer 3 Report

Comments and Suggestions for Authors

No more comments